# Sophoraflavanone G from *Sophora flavescens* Ameliorates Allergic Airway Inflammation by Suppressing Th2 Response and Oxidative Stress in a Murine Asthma Model

**DOI:** 10.3390/ijms23116104

**Published:** 2022-05-29

**Authors:** Meng-Chun Wang, Wen-Chung Huang, Li-Chen Chen, Kuo-Wei Yeh, Chwan-Fwu Lin, Chian-Jiun Liou

**Affiliations:** 1Department of Traditional Chinese Medicine, Chang Gung Memorial Hospital, Taoyuan 33378, Taiwan; mengchun1023@gmail.com; 2Graduate Institute of Health Industry Technology, Research Center for Food and Cosmetic Safety, Chang Gung University of Science and Technology, Taoyuan 33303, Taiwan; wchuang@mail.cgust.edu.tw; 3Division of Allergy, Asthma, and Rheumatology, Department of Pediatrics, Chang Gung Memorial Hospital, Linkou, Taoyuan 33305, Taiwan; lcchen@cgmh.org.tw (L.-C.C.); kwyeh@cgmh.org.tw (K.-W.Y.); 4Department of Pediatrics, New Taipei Municipal TuCheng Hospital (Built and Operated by Chang Gung Medical Foundation), New Taipei 23656, Taiwan; 5Department of Cosmetic Science, Research Center for Chinese Herbal Medicine, Chang Gung University of Science and Technology, Taoyuan 33303, Taiwan; 6Department of Anesthesiology, Chang Gung Memorial Hospital, Linkou, Taoyuan 33305, Taiwan; 7Department of Nursing, Division of Basic Medical Sciences, Research Center for Chinese Herbal Medicine, Chang Gung University of Science and Technology, Taoyuan 33303, Taiwan

**Keywords:** airway hyper-responsiveness, airway inflammation, asthma, sophoraflavanone G, Th2 cell

## Abstract

Sophoraflavanone G (SG), isolated from *Sophora flavescens*, has anti-inflammatory and anti-tumor bioactive properties. We previously showed that SG promotes apoptosis in human breast cancer cells and leukemia cells and reduces the inflammatory response in lipopolysaccharide-stimulated macrophages. We investigated whether SG attenuates airway hyper-responsiveness (AHR) and airway inflammation in asthmatic mice. We also assessed its effects on the anti-inflammatory response in human tracheal epithelial cells. Female BALB/c mice were sensitized with ovalbumin, and asthmatic mice were treated with SG by intraperitoneal injection. We also exposed human bronchial epithelial BEAS-2B cells to different concentrations of SG to evaluate its effects on inflammatory cytokine levels. SG treatment significantly reduced AHR, eosinophil infiltration, goblet cell hyperplasia, and airway inflammation in the lungs of asthmatic mice. In the lungs of ovalbumin-sensitized mice, SG significantly promoted superoxide dismutase and glutathione expression and attenuated malondialdehyde levels. SG also suppressed levels of Th2 cytokines and chemokines in lung and bronchoalveolar lavage samples. In addition, we confirmed that SG decreased pro-inflammatory cytokine, chemokine, and eotaxin expression in inflammatory BEAS-2B cells. Taken together, our data demonstrate that SG shows potential as an immunomodulator that can improve asthma symptoms by decreasing airway-inflammation-related oxidative stress.

## 1. Introduction

The global prevalence of asthma, a chronic allergic respiratory disease, is gradually increasing [1]. Allergen inhalation or microbial infections can induce the onset of asthma. In recent years, climate change has expanded allergen exposure, and severe air pollution has boosted the incidence of respiratory conditions such as asthma and chronic obstructive pulmonary disease. An acute asthma attack causes chest tightness, rapid coughing, dry coughing, and difficulty breathing [2]. Repeated asthma attacks, however, can cause the smooth muscles of the airway to thicken, and their contraction in asthmatic attacks will narrow the airway [3]. In addition, epithelial cells release more mucus into the obstructed airway. If a patient with an acute asthma attack does not urgently use a bronchodilator to relieve this constriction, the patient will have difficulty breathing or even suffocate to death [1].

Asthma can be divided into eosinophilic asthma and non-eosinophilic asthma according to the distribution of immune cells in bronchoalveolar lavage fluid (BALF) or sputum. Non-eosinophilic asthma, considered to be non-type 2 asthma, can be categorized into three types: neutrophilic asthma, mixed granulocytic asthma, and paucigranulocytic asthma [4]. Induced-sputum neutrophil and BALF were higher in patients with smoking asthma, and treatment with inhaled steroids is ineffective in these patients [5].

In recent years, studies have shown excessive eosinophil infiltration in the lungs of patients with asthma. Inflammatory eosinophils release inflammation mediators, causing severe inflammation and an allergic response in the lungs [6]. Moreover, the lungs of asthmatic patients will exhibit airway hyper-responsiveness (AHR) and airway remodeling that can induce tracheal goblet cell hyperplasia and increased mucus secretions [7]. Previous studies have found that the pathological characteristics of asthma and airway remodeling are closely related to the excessive activation of Th2 cells. Activated Th2 cells will release a large amount of the cytokines interleukin (IL)-4, IL-5, and IL-13, not only causing an imbalance in Th2/Th1 cells in the immune system but also aggravating immunoglobulin (Ig)E production by B cells to induce mast cell activation [8]. Therefore, regulating Th2 cell activation could ameliorate the development of severe asthma symptoms.

Furthermore, allergens also stimulate pulmonary alveolar epithelial cells and tracheal epithelial cells to release reactive oxygen species (ROS), which can cause oxidative damage to lung tissue [9]. Oxidative stress can stimulate tracheal epithelial cells to secrete more mucus into obstructed airways and induce contraction of airway smooth muscles, leading to shortness of breath and difficulty breathing during an asthma attack [10]. Previous studies have shown that ROS and antioxidants can affect the mitochondrial function in the developing airway smooth muscle [11]. The mitochondrial fatty acid beta-oxidation of bronchial smooth muscle could exacerbate airway remodeling in asthma [12].

Herbal medicine has been widely used in China and other parts of Asia for thousands of years. Chinese herbal complex formulas, including Ding Chuan Tang and Xiao-Qing-Long-Tang, have shown potential to relieve asthma symptoms [13]. Some pure compounds of Chinese herbal medicines may improve asthma, possibly by regulating specific inflammation or immune cell targets, and offer a straightforward way to investigate molecular mechanisms [14]. In traditional Chinese medicine, the root of *Sophora flavescens* is used to treat liver inflammation, diuresis, and swelling [15]. Previous studies found that *S. flavescens* could attenuate granulomatous inflammation during *Mycobacterium tuberculosis* infection [16]. The ethyl acetate extracts of *S. flavescens* could also reduce the expression of pro-inflammatory cytokines and mediators in lipopolysaccharide-stimulated mouse macrophages [17]. Several pure compounds have been isolated from the roots of S. flavescens, including matrine, oxymatrine, kushenol, kuraridine, kurarinone, and sophoraflavanone G (SG) [18]. Kurarinone has an anti-inflammatory and antioxidant effect for improving rheumatoid arthritis in mice [19]. Matrine ameliorated lung injury in mice and reduced the inflammatory response in lung epithelial cells [20]. Oxymatrine reduced ROS production in RANKL-induced osteoclast formation by suppressing SREBP2 and NFATc1 expression [21]. Kushenol C isolated from the roots of *S. flavescens* could prevent ROS production in tert-butyl hydroperoxide-induced HaCaT keratinocytes [22]. We previously showed that matrine can improve airway inflammation and eosinophil infiltration in asthmatic mice [23], and that SG reduces the inflammatory response in lipopolysaccharide-stimulated macrophages [24]. SG could inhibit the neuroinflammation in lipopolysaccharide-activated microglia through regulated MAPKs and Nrf2/HO-1 signaling pathways [25]. In the current work, we investigated the ability of SG to improve AHR, airway inflammation, and oxidative stress, and evaluated its effects on immune regulation function in asthmatic mice.

## 2. Results

### 2.1. SG Attenuates AHR in Mice

The experimental protocol for the sensitized asthma mouse model is shown in Figure 1A.

First, we evaluated whether SG ameliorates abnormal airflow of the respiratory tract in asthmatic mice. To measure AHR and evaluate lung function in mice, we exposed the animals to aerosolized methacholine. With inhalation of 40 mg/mL methacholine, OVA-induced asthmatic mice exhibited markedly elevated enhanced pause values compared with unsensitized control mice. Our results demonstrated that SG could effectively attenuate enhanced pause values when compared with the OVA mice (OVA, 7.97 ± 1.56 vs. SG5, 6.12 ± 0.94 (*p* < 0.05) and SG10, 4.06 ± 0.81 (*p* < 0.01)) (Figure 1B). 

### 2.2. SG Reduces Eosinophil Numbers in BALF

We calculated the number of inflammatory cells in the BALF to investigate whether SG reduced the inflammatory response in asthmatic mice. Inflammatory cells were stained with Giemsa stain. We detected a larger number of eosinophils in BALF from the OVA group compared with the unsensitized control mice. In addition, SG10 decreased eosinophils and total cell number in BALF of OVA-sensitized mice compared with controls (Figure 1C).

### 2.3. SG Reduces Eosinophil Infiltration and Goblet Cell Hyperplasia in Murine Lung

In the lungs of asthmatic patients, a large infiltration of activated eosinophils could cause an allergic and inflammatory response [26]. Asthmatic mice treated with SG had reduced eosinophil infiltration of the lungs compared with their OVA-sensitized untreated counterparts (Figure 2A,B). Next, a lung biopsy was stained with PAS solution to observe goblet cell hyperplasia in the trachea of mice [27]. The PAS staining results showed remarkably upregulated tracheal goblet cell hyperplasia in the OVA-sensitized group. SG-treated animals, however, had notably reduced hyperplasia compared with their OVA-sensitized untreated counterparts (Figure 3A,B).

### 2.4. SG Regulates Cytokine and Chemokine Expression in BALF and Lung Tissue 

Asthma is an allergic disease of immune imbalance, and overactivation of Th2 cells is induced in the lung to exacerbate the secretion of IL-4, IL-5, and IL-13 cytokines [28]. In BALF, SG significantly decreased IL-4, IL-5, IL-13, TNF-α, IL-6, CCL11, and CCL24 levels in comparison with OVA-induced asthmatic mice (Figure 4). Real-time PCR was applied to assess relative mRNA expression in murine lung tissue, with results indicating reduced IL-4, IL-5, IL-13, TNF-α, IL-6, CCL11, and CCL24 gene expression in SG-treated asthmatic mice compared with the OVA-only control group (Figure 5). Furthermore, in BALF and lung tissue, SG was associated with significantly elevated interferon (IFN)-γ expression compared with the OVA group (Figure 4D and Figure 5D). 

### 2.5. SG Modulates GSH, SOD, and MDA Expression in the Lungs

Continuous asthma attacks will stimulate oxidative stress in the lung and weaken lung function [29]. Previous studies demonstrated that GSH, CAT, and SOD have anti-oxidative stress effects, reducing lung damage in patients with asthma [30]. Compared with untreated OVA-sensitized mice, asthmatic mice treated with SG had elevated GSH, CAT, and SOD activity in lung tissue (Figure 6A–C). Treatment with SG also notably reduced MDA levels compared with levels in the OVA-sensitized mice (Figure 6D).

### 2.6. SG Inhibits Levels of Serum OVA-Specific IgG1 and IgE 

Activated Th2 cells produce IL-4, which is responsible for IgE production by B cells to activate eosinophils and mast cells [31]. We used ELISA to evaluate the effect of SG on serum levels of OVA-specific antibodies. The OVA group had the highest levels of OVA-IgG1 and OVA-IgE, and SG-treated OVA-sensitized mice had significantly decreased OVA-IgG1 and OVA-IgE levels compared with the OVA group (Figure 7A,B). Furthermore, OVA-IgG2a levels were significantly elevated in the SG10 groups compared with levels in the OVA-sensitized animals (Figure 7).

### 2.7. SG Inhibits Th2-Associated Cytokine Expression in Splenocytes

Splenocytes were cultured with 100 µg/mL OVA for 5 continuous days. In the OVA group, splenocytes showed increased IL-4, IL-5, and IL-13 secretions relative to splenocytes from normal mice. However, splenocytes from asthmatic mice treated with SG had significantly decreased IL-4, IL-5, and IL-13 levels compared with the OVA animals (Figure 8A–C). Furthermore, SG was associated with increased IFN-γ production compared with the OVA group (Figure 8D).

### 2.8. SG Reduces the Inflammatory Response in BEAS-2B Cells 

After BEAS-2B cells were treated with SG and with 10 ng/mL TNF-α for 24 h, SG-treated cells had significantly decreased levels of CCL5, MCP-1, IL-8, and IL-6 compared with untreated, activated BEAS-2B cells (Figure 9A–D). Levels of CCL11 and CCL24 also were significantly decreased with SG treatment in IL-4/TNF-α-activated BEAS-2B cells (Figure 9E,F).

## 3. Discussion

*S. flavescens* is a plant mainly grown in China and other Asian countries [32]. Traditional Chinese medicine relies on *S. flavescens* to treat liver inflammation, jaundice, and fever [15]. Complex formulas containing *S. flavescens* also have been found to improve asthma and atopic dermatitis [32]. Earlier studies have shown that an anti-asthma herbal medicine intervention (containing *Ganoderma lucidum*, *S. flavescens*, and *Glycyrrhiza uralensis*) can effectively reduce AHR, airway smooth muscle contraction, and activation of Th2 cells in asthmatic mice [33]. According to these previous studies, *S. flavescens* may contain active ingredients that can improve asthma symptoms [33]. We have found that SG inhibits the inflammatory response in LPS-activated macrophages [24], which led us to speculate that SG might dampen airway inflammation in asthmatic mice. The results of the current study suggest that SG reduced AHR, eosinophil infiltration, and goblet cell hyperplasia in OVA-sensitized mice. SG also reduced Th2-associated cytokine and eotaxin expression in BALF in the lung and decreased OVA-specific IgE in serum of OVA-induced allergic asthmatic mice.

AHR is an important indicator of lung function, mainly for detecting airway flow and respiratory rate, and can be used to assess respiratory system function in patients with asthma [34]. In patients with chronic asthma, because of repeated asthma attacks, airway elasticity is reduced, and airway contraction force is attenuated. Moreover, alveolar surface tension is reduced in asthmatic patients, resulting in a sudden asthma attack that interferes with inhalation and produces shortness of breath [3]. Our findings are that SG reduced AHR in asthmatic mice and could improve shortness of breath and respiratory function. 

Higher IL-13 levels have been detected in BALF and lung tissue of patients with asthma, and previous studies have confirmed that excessive IL-13 secretion from Th2 cells deteriorates respiratory function [31]. Mice with asthma induced by house dust mites and treated with anti-IL-13 therapy show reduced AHR [35], and AHR cannot be triggered in knockout IL-13 mice with induced asthma [36]. Our experiments confirmed that SG-treated asthmatic mice can effectively show reduced IL-13 expression in the lungs and BALF, along with inhibited IL-13 from spleen cells. These results indicate that SG could slow AHR in asthmatic mice and restore lung function through inhibition of IL-13 expression.

Goblet cell hyperplasia is a characteristic of airway remodeling in patients with asthma. Allergens that chronically stimulate airway epithelial cells may induce their differentiation into goblet cells [37]. As noted, activated goblet cells secrete excessive mucus, causing airway obstruction during an asthma attack and aggravating breathing difficulties, even to the point of suffocation and death [38]. For these reasons, reducing tracheal goblet cell hyperplasia is an important strategy for improving the pathological symptoms of asthma. IL-13 and IL-4 are important cytokines in goblet cell hyperplasia induction [37], and treatment with anti-IL-13 or anti-IL-4 suppresses this hyperplasia in asthmatic mice [31]. Because PAS mainly stains glycoproteins, which drive the activation of goblet cells in tracheal epithelium and mucus secretion [39], we used this staining to detect tracheal goblet cell hyperplasia in lung sections. We found that SG treatment significantly reduced tracheal goblet cell hyperplasia in asthmatic mice. SG treatment in asthmatic mice thus appears to effectively reduce IL-4 and IL-13 expression in lung, BALF, and spleen cells, implying reduced mucus hypersecretion in the airway. 

Asthma with eosinophil infiltration is the most common type of asthma [40]. Excessive IL-5 secretion by activated Th2 cells stimulates cell differentiation into more mature eosinophils in the bone marrow and induces greater proliferation of activated eosinophils [41]. IL-4 and TNF-α can stimulate tracheal epithelial cells to release more eotaxins (CCL11, CCL24, and CCL26) [42], and IL-4 can stimulate lung epithelial cells to release high amounts of CCL26 [43]. In this way, the lungs and tracheal epithelial cells release a large amount of eotaxins, which attract mature eosinophil migration into the lungs, leading to high eosinophil infiltration. Activated eosinophils release inflammation-related molecules and induce oxidation proteins, including major eosinophil cationic proteins and basic proteins, which cause inflammation and oxidative damage to alveolar cells. These proteins also cause tracheal epithelial cell inflammation and stimulate goblet cell activation to release more mucus, obstructing the airway [44]. In the current study, lung sections from OVA-induced asthmatic mice also showed high eosinophil infiltration, and eosinophils in BALF samples were significantly higher than in normal mice. Of note, SG treatment significantly reduced the number of eosinophils in the lungs and BALF. SG can inhibit Th2 and eosinophil activation and IL-5 release. In addition, SG reduced lung inflammation and suppressed activation of pulmonary macrophages to release more TNF-α. Therefore, SG reduced IL-4/TNF-α release from immune cells and the subsequent stimulation of tracheal epithelial cell activation and eotaxin secretion. As SG likely reduced IL-5 release by Th2 cells and blocked eosinophil differentiation, this plant-derived compound may improve lung inflammation and oxidative damage.

Asthma is an allergic disease, and patients with allergies have high serum IgE. In asthma patients, Th2 cells secrete more IL-4 to induce B cell activation, leading to excessive IgE secretion [8]. When lung mast cells in asthmatic patients combine with IgE and allergens, they are activated to release a large amount of histamine and inflammatory mediators. This release causes allergic and inflammatory reactions in the lungs, increasing lung cell damage and reducing lung function [31]. In this study, SG significantly reduced serum OVA-IgE and contributed to regulation of the immune system. As noted, asthma is also a manifestation of excessive activation of Th2 cells, which leads to increased secretion of IL-4, IL-5, and IL-13 in asthmatic mice [31]. This activation inhibits Th1 cell activation and reduces IFN-γ secretion. Here, we confirmed that SG could reduce levels of OVA-IgG1 (a subtype of Th2 cells) and increase OVA-IgG2a (a subtype of Th1 cells) levels in the serum of asthmatic mice. These findings confirm that SG indeed can improve immunomodulation in this asthma model and reduce excessive Th2 cell activation. 

The lungs of patients with asthma are in a state of inflammation, showing high levels of TNF-α and IL-6 in lung tissue and BALF [45], mainly released by activated macrophages [31]. The continuous inflammation also causes collagen deposition in alveolar cells, leading to pulmonary fibrosis [46]. SG also reduced the levels of IL-6 and TNF-α in BALF and lung tissue of asthmatic mice. These inflammatory cytokines can stimulate tracheal epithelial cells to secret more inflammatory cytokines, which damages the physiological function of lung tissue. Therefore, we thought that SG inhibited IL-6 production of inflammatory BEAS-2B cells, which could contribute to attenuate pulmonary inflammation in asthmatic mice. In addition, these inflammatory cytokines can stimulate tracheal epithelial cell activation, leading to more chemokines (MCP-1, CCL5, and IL-8), attracting macrophage and neutrophil migration into the lung and aggravating oxidative stress and cell damage [47]. Our experiments confirmed that SG treatment leads to reduced expression of inflammatory cytokines in the lungs and BALF of asthmatic mice and inhibits the release of more inflammatory cytokines and chemokines in activated tracheal epithelial cells. In this way, SG can dampen the inflammatory airway response in asthmatic mice. 

Activated immune cells and inflamed tracheal epithelial cells will release oxidative molecules and induce oxidative stress that damages the lung cells in patients with asthma [38]. Oxidative stress not only increases proliferation and contraction of tracheal smooth muscle but also stimulates mucus secretions from tracheal epithelial cells [48]. Continuous oxidative stress can also induce apoptosis and DNA damage in alveolar and tracheal epithelial cells [47]. As a result, lung function of asthmatic patients will continue to deteriorate, exacerbating dyspnea and suffocation. In the current work, we found that SG significantly increased expression of SOD, CAT, and GSH and reduced MDA production in the lungs of asthmatic mice. Superoxide is an important ROS that can kill bacteria and cause cell damage [9]. SOD could catalyze superoxide radicals to form hydrogen peroxide, which has a bactericidal effect and also can cause oxidative cell damage [29]. CAT can decompose H_2_O_2_ to produce water and oxygen, reducing cellular damage [9]. GSH is another antioxidant that can eliminate ROS and reduce lung cell damage [10], and MDA is an important indicator of cellular oxidative stress, as both lipid metabolism and peroxidation will increase its production [29]. Our results suggest that SG is a natural compound that effectively reduces oxidative stress and related damage in asthmatic lungs.

## 4. Materials and Methods

### 4.1. Materials

SG (purity ≥98% by HPLC) was purchased from ChemFaces (Wuhan, China). For cell experiments, SG was dissolved in dimethylsulfoxide (DMSO) in a stock solution of 30 mM SG. For animal experiments, SG was formulated as 5 mg/50 μL and 10 mg/50 μL in DMSO.

### 4.2. Mouse Sensitization and Administration of SG

Female BALB/c mice, age 6–8 weeks, were purchased from the National Laboratory Animal Center (Taipei, Taiwan). Mice were housed in a temperature-controlled animal room, under a 12 h light/dark cycle, and allowed free access to standard chow diet and water. The animal experiments were approved by the Animal Care and Protection Committee of Chang Gung University of Science and Technology (approval number: 2018-004). The experimental protocol for the sensitized asthma mouse model is shown in Figure 1A. All mice were randomly divided into four groups (*n* = 10 in each group): a normal control (N group), treated with nothing; an ovalbumin (OVA)-sensitized control (OVA group; OVA from Sigma Aldrich, St. Louis, MO, USA); and an OVA-sensitized group treated with 5 mg/kg or 10 mg/kg SG (SG5 and SG10 groups, respectively). Mice were sensitized with a solution containing 50 μg OVA and 0.8 mg AlOH_3_ adjuvant in 200 μL normal saline, administered by intraperitoneal injection on days 1–3 and day 14. The lung allergy challenge was triggered with 2% atomized OVA by a compressor nebulizer (Medical Depot, Inc., Port Washington, NY, USA) on days 14, 17, 20, 23, and 27. Mice were administered SG or DMSO (as negative control) by intraperitoneal injection 1 h before OVA challenge or methacholine inhalation (see below). On day 28, mice were evaluated for AHR, as described below. On day 29, the animals were sacrificed to allow for investigation of the pathological features of asthma, inflammatory airway, and immunomodulatory effects.

### 4.3. Lung Function Analysis

AHR was measured 24 h after the last OVA challenge, as described previously [49]. Briefly, mice were placed in a closed system to inhale aerosolized methacholine (0 to 40 mg/mL) for 3 min. Subsequently, the animals were placed in a closed chamber for detection of the enhanced pause data using a whole-body plethysmograph system (Buxco Electronics, Troy, NY, USA).

### 4.4. Serum Collection and Splenocyte Culture

Animals first were placed in an anesthesia box with 4% isoflurane to induce adequate anesthesia. Immediately afterwards, blood was taken from the orbital vascular plexus and centrifuged at 6000 rpm to collect serum for OVA-specific antibody detection. The supernatant was obtained for detection of cytokine concentrations as described previously [50]. Furthermore, spleens were removed, and splenocytes (5 × 10^6^ cells/mL) were seeded on culture plates containing 100 µg/mL OVA solution for 5 days. 

### 4.5. Bronchoalveolar Lavage Fluid (BALF)

After animals were anesthetized and killed, an indwelling needle was intubated into the trachea and lungs washed with 1 mL sterile normal saline. The collected wash solution was defined as BALF, which we used to detect chemokines and cytokines by ELISA. BALF cells were stained using Giemsa stain (Sigma) for identification and counting of different types of immune cells. 

### 4.6. ELISA 

Chemokine and cytokine levels were detected in splenocyte culture medium and BALF using ELISA kits (R&D Systems, Minneapolis, MN, USA). For detection of OVA-IgE levels, serum was diluted 5-fold and results reported as OD_450_ values. Standard curves for OVA-IgG2a and OVA-IgG1 were generated based on values from pooled serum from OVA-sensitized mice. All serum OVA-specific antibodies were detected using the respective specific ELISA kit (BD Biosciences, Franklin Lakes, NJ, USA).

### 4.7. Glutathione (GSH), Catalase (CAT), and Superoxide Dismutase (SOD) Assay

Lung tissues were homogenized using a homogenizer (FastPrep-24, MP Biomedicals, Santa Ana, CA, USA). We used specific assay kits for CAT, SOD, and GSH (Sigma) to assay levels in the lung tissues according to the manufacturer’s instructions. CAT, SOD, and GSH levels were detected on a multi-mode microplate reader (SpectraMax i3X, Molecular Devices, San Jose, CA, USA).

### 4.8. Malondialdehyde (MDA) Assay

Lung tissues were homogenized, and a lipid peroxidation assay kit was used to detect MDA levels (Abcam, Waltham, MA, USA) as described previously [49]. Briefly, a lung tissue solution was incubated with the MDA color reagent solution, followed by incubation with reaction solution for 30 min. MDA levels were detected at 695 nm using a microplate reader (Multiskan FC, Thermo, Waltham, MA, USA).

### 4.9. Lung Tissue Histopathology 

Lungs were fixed and embedded in paraffin, and 6 μm sections were prepared. Lung tissues were stained with hematoxylin and eosin (HE, Sigma-Aldrich) for evaluation of eosinophil infiltration and with periodic acid–Schiff (PAS) solution (PAS; Sigma) for assessment of tracheal goblet cell hyperplasia, as described previously [51].

### 4.10. Real-Time PCR Analysis 

Lung tissues were homogenized, and RNA was extracted using TRIzol reagent solution (Life Technologies, Carlsbad, CA, USA). Subsequently, RNA was reverse transcribed into cDNA using a cDNA synthesis kit, and gene expression was examined using a real-time PCR system (iCycler; Bio-Rad, San Francisco, CA, USA), as described previously [27]. 

### 4.11. SG Treatment of BEAS-2B Cells 

Human bronchial epithelial cells (BEAS-2B) (American Type Culture Collection, Manassas, VA, USA) were seeded at a density of 2 × 10^5^ cells on a 24-well plate with DMEM/F12 medium and incubated with SG (0–30 μM) for 1 h, followed by stimulation with 10 ng/mL tumor-necrosis factor (TNF)-α for 24 h. The supernatants were collected for detection of chemokine or cytokine production using the respective ELISA kits [52]. BEAS-2B cells treated with SG also were stimulated with 10 ng/mL IL-4/TNF-α for 24 h for analysis of eotaxin (CCL11 and CCL24) secretions. BEAS-2B cells used passages 6–8 for cell experiment.

### 4.12. Statistical Analysis

All experiments were repeated at least three times. Statistical analysis was performed with one-way ANOVA followed by the Tukey–Kramer post hoc test. The values are shown as the mean ± SEM, and *p* < 0.05 was considered statistically significant.

## 5. Conclusions

Our findings collectively show that SG treatment significantly reduced AHR, eosinophil infiltration, goblet cell hyperplasia, and airway inflammation in the lungs of asthmatic mice. SG inhibited oxidative stress in the lungs of asthmatic mice. SG also suppressed levels of Th2 cytokines and chemokines in BALF and lung tissue. We confirmed that SG decreased pro-inflammatory cytokine, chemokine, and eotaxin expression in inflammatory BEAS-2B cells. In conclusion, we suggest that SG has extremely potential pharmacologic benefits as an immunomodulator in reducing anti-oxidative stress and inflammation in asthma.

## Figures and Tables

**Figure 1 ijms-23-06104-f001:**
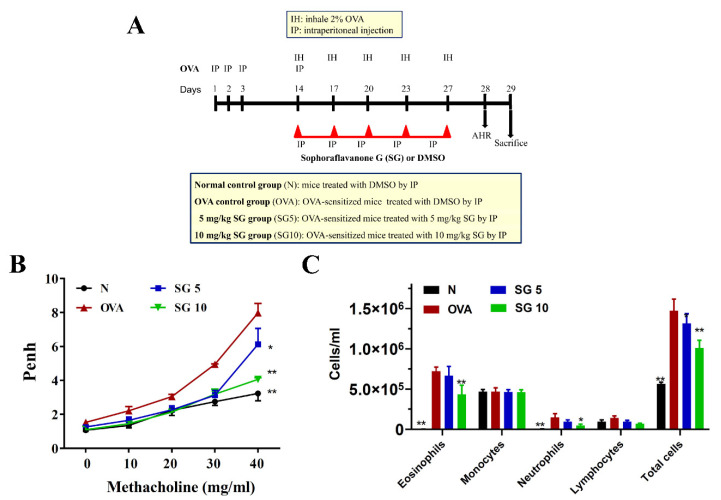
Effect of sophoraflavanone G (SG) on pulmonary function in asthmatic mice. (**A**) Experimental procedures for asthmatic mouse studies. (**B**) Enhanced pauses (Penh) were detected by methacholine inhalation (0–40 mg/mL) in mice. (**C**) Numbers of inflammatory and total cells in BALF. The data are presented as mean ± SEM of three independent experiments (*n* = 10 per group); * *p* < 0.05, ** *p* < 0.01 versus OVA group.

**Figure 2 ijms-23-06104-f002:**
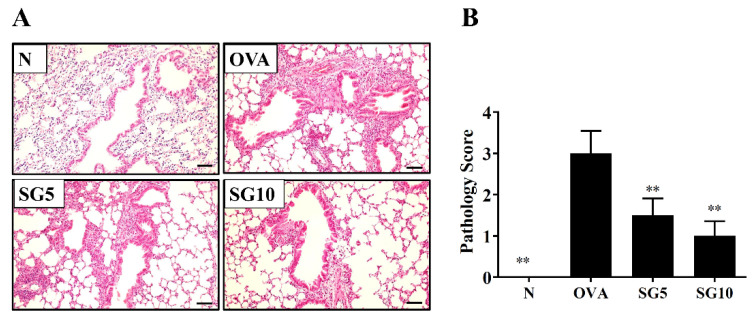
Sophoraflavanone G (SG) reduces eosinophil infiltration in the mouse lung. (**A**) H&E staining showing eosinophil infiltration (200× magnification). (**B**) Inflammatory scores are for lung sections. The data are presented as mean ± SEM of three independent experiments (*n* = 4–6 per group); ** *p* < 0.01 versus the OVA group (scale bar = 100 µm).

**Figure 3 ijms-23-06104-f003:**
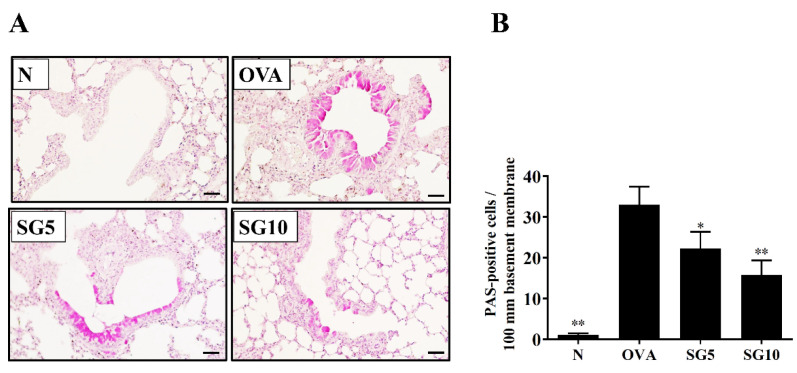
Sophoraflavanone G (SG) reduces goblet cell hyperplasia in the mouse lung. (**A**) PAS staining showing goblet cell hyperplasia (200× magnification). (**B**) PAS-positive cells were calculated per 100 μm in the trachea. The data are presented as mean ± SEM of three independent experiments (*n* = 4–6 per group); * *p* < 0.05, ** *p* < 0.01 versus the OVA group (scale bar = 100 µm).

**Figure 4 ijms-23-06104-f004:**
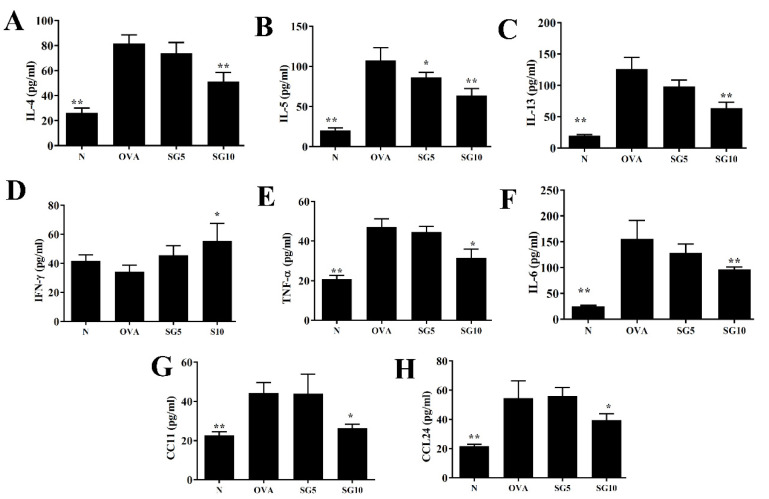
Sophoraflavanone G (SG) regulates BALF cytokine and chemokine levels. (**A**) IL-4, (**B**) IL-5, (**C**) IL-13, (**D**) IFN-γ, (**E**) TNF-α, (**F**) IL-6, (**G**) CCL11, and (**H**) CCL24 as measured by ELISA. The data are presented as mean ± SEM of three independent experiments (*n* = 10 per group); * *p* < 0.05, ** *p* < 0.01 versus the OVA group.

**Figure 5 ijms-23-06104-f005:**
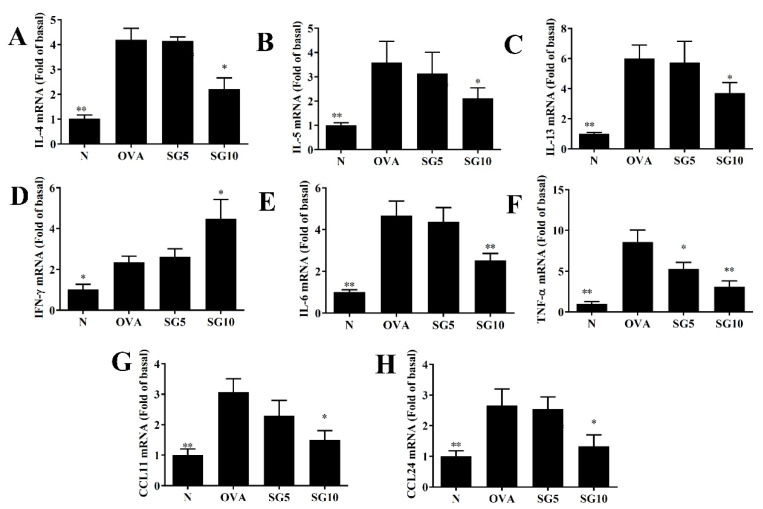
Sophoraflavanone G (SG) regulates cytokine and chemokine gene expression in the lung. (**A**) IL-4, (**B**) IL-5, (**C**) IL-13, (**D**) IFN-γ, (**E**) IL-6, (**F**) TNF-α, (**G**) CCL11, and (**H**) CCL24 were determined by real-time PCR. Fold values relative to β-actin expression. The data are presented as mean ± SEM of three independent experiments (*n* = 10 per group); * *p* < 0.05, ** *p* < 0.01 versus the OVA group.

**Figure 6 ijms-23-06104-f006:**
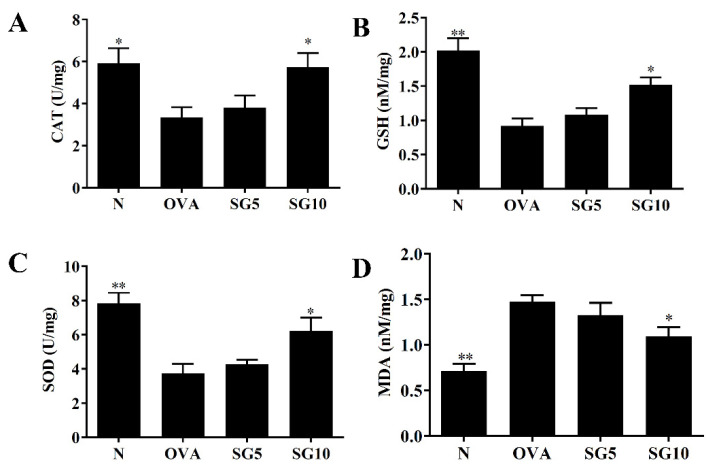
Sophoraflavanone G (SG) regulates oxidative stress factors. (**A**) CAT, (**B**) GSH, (**C**) SOD, and (**D**) MDA were measured in lung tissues. The data are presented as mean ± SEM of three independent experiments (*n* = 10 per group); * *p* < 0.05, ** *p* < 0.01 versus the OVA group.

**Figure 7 ijms-23-06104-f007:**
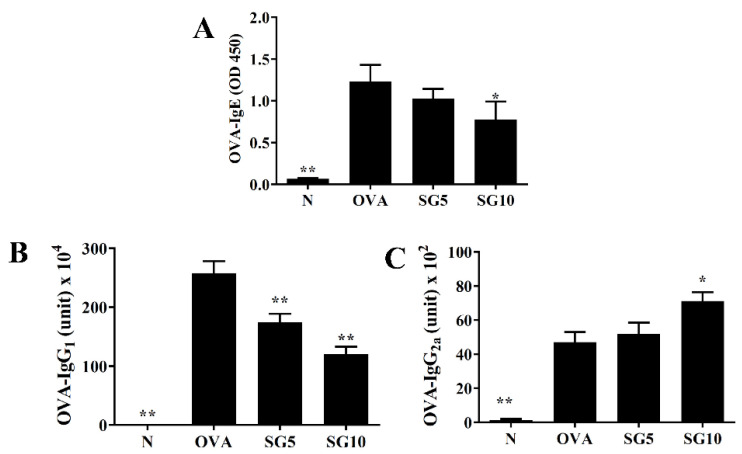
Sophoraflavanone G (SG) regulated OVA-specific antibodies in serum. (**A**) OVA-IgE, (**B**) OVA-IgG_1_, and (**C**) OVA-IgG_2a_ were detected by ELISA. The data are presented as mean ± SEM of three independent experiments (*n* = 10 per group); * *p* < 0.05, ** *p* < 0.01 versus the OVA group.

**Figure 8 ijms-23-06104-f008:**
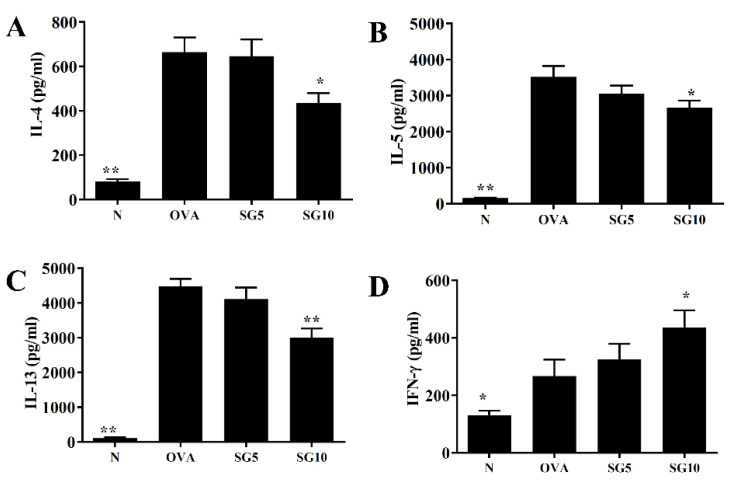
Sophoraflavanone G (SG) regulates cytokine production in OVA-activated splenocytes. (A) IL-4, (**B**) IL-5, (**C**) IL-13, and (**D**) IFN-γ levels were examined by ELISA. The data are presented as mean ± SEM of three independent experiments (*n* = 10 per group); * *p* < 0.05, ** *p* < 0.01 versus the OVA group.

**Figure 9 ijms-23-06104-f009:**
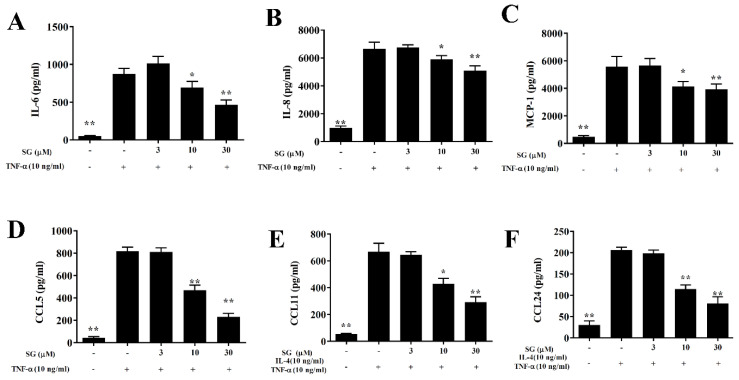
Sophoraflavanone G (SG) reduces inflammatory cytokine and chemokine production in BEAS-2B cells. BEAS-2B cells (6–8 passages) were treated with SG, then stimulated with TNF-α (10 ng/mL) for 24 h. (**A**) IL-6, (**B**) IL-8, (**C**) MCP-1, and (**D**) CCL5 were examined by ELISA. Values are mean ± SEM; * *p* < 0.05, ** *p* < 0.01 versus BEAS-2B cells stimulated with TNF-α alone. (**E**,**F**) Cells also were treated with SG and then stimulated with 10 ng/mL TNF-α/IL-4 for 24 h. (**E**) CCL11 and (**F**) CCL24 were detected. The data are presented as mean ± SEM of three independent experiments (*n* = 12 per group); * *p* < 0.05, ** *p* < 0.01 versus BEAS-2B cells stimulated with TNF-α /IL-4.

## Data Availability

The data presented in this study are available on request from the corresponding author.

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
