# Peer review of "Sophoraflavanone G from Sophora flavescens Ameliorates Allergic Airway Inflammation by Suppressing Th2 Response and Oxidative Stress in a Murine Asthma Model"

_ijms, 2022, doi:10.3390/ijms23116104_

Round 1

Reviewer 1 Report

The "Sophoraflavanone G from Sophora flavescens ameliorates allergic airway
inflammation by suppressing Th2 response and oxidative stress in a murine
asthma model" manuscript makes important contributions to research on reducing the processes of oxidative stress and inflammation associated with asthma.

Introduction

Please offer more details regarding oxidative stress induce contraction of airways smooth muscle - line 72.

Please offer also more detail about Sophora flavescens effects on other inflammatory disorders.

Based on their results, conclusions should contain more explanations regarding the efficacy of Sophora flavescens in their experimental study.

Congratulation for your work!

Author Response

The "Sophoraflavanone G from Sophora flavescens ameliorates allergic airway

inflammation by suppressing Th2 response and oxidative stress in a murine

asthma model" manuscript makes important contributions to research on reducing the processes of oxidative stress and inflammation associated with asthma.

Introduction

Please offer more details regarding oxidative stress induce contraction of airways smooth muscle - line 72.

Responses:

Thank for reviewer’s suggestion. We add some oxidative stress effect airways smooth muscle in Introduction (Line 75-80)

Please offer also more detail about Sophora flavescens effects on other inflammatory disorders.

Thank for reviewer’s suggestion. We add more Sophora flavescens or pure compound to improve inflammatory response in the Introduction (Line 87-102)

Based on their results, conclusions should contain more explanations regarding the efficacy of Sophora flavescens in their experimental study.

Responses:

We thank the reviewer for pointing this out. We modify some sentences in conclusions.

Congratulation for your work!

Responses:

Thank you for recognising my work. I am grateful to hear that I am on the right track.

Reviewer 2 Report

1Wang and colleagues present a mostly murine-based study on how Sophoraflavone G influences the clincial and molecular phenotype in an asthma mouse model. The autors undetook sereval bioassays to characterise the breathing properties of mice and the immunological phenoytpe of organ compartments. However, even though this is an impressive amount of data, the reviewer feels that the manuscript needs to be improved to make the findings acessible for the readership of The Journal. In detail:      

11.       Introduction: Asthma is characterised by multiple endophenotypes. One distinction is “eosinophilic asthma” vs “non-eosinophilic asthma”. In their introduction, the authors focus on   esosinic & Th2-mediated subtype only, for the sake of completeness, please add the information for the readers from the non-asthma-field on how brouad the cinicla spectrum of asthma is on the molecular level as the Int. J. Mol. Sci. is a journal that attracts an audience from several fields.

22.       Introductory paragraph on ROS in asthma: appears unlinked to text surrounding it, might be better placed in context (How do ROS and SG interrelate?) or removed.

33.       Figure 1, 2, 3, 4, 5, 6, 7 and 8: please add no of observations (independent technical and biological replicates, i.e. no of measurements, no of litters etc) and figure 9: please add number of technical and biological replicates (passages) for the experiments on BAS2B. This is especially of importance as the current manuscript focusses on “significant is this, hence discussed is this” while some information might lie in those observations for which no significant difference was observed (“the dog that did not bark”). Thus, it is vital to discriminate “not significant because the effect of the intervention is absent or too small to be detected” versus “not significant because the number of independent biological replicates is very small”.

44.       Text affiliated with figures 1 to 8: it would be helpful for the reader to understand the rationale why certain things have been measured (and others have not). Please again keep in mind that readers of The Journal come from different academic backgrounds beyond asthma, so that it is not self-explanatory why IL13 has been investigated for figure 8 while, for instance,  IL10 has not. This does not imply that the reviewer thinks data on IL10 needs to be provide, only the authors choice of biomarkers needs to be more transparent throughout the manuscript’s results section. This is important as there are technical reasons for restriction in some cases: individual transcripts or proteins were targeted (researchers choice) versus a pre-build screening set provided by a company was targeted (company’s choice).

55.       Text affiliated with Figure 9: there is certainly a well-thought-out reason why BEAS2B cells were included in this manuscript that entirely deals with murine models before. Please state clearly what the benefit of this experiment is.

66.       Figure 9: the six subpanels A to F show a very similar appearance: one small bar, two large bars, two smaller bars. This is independent of the intervention (IL10, TNFalpha) and independent of the biomarker anlysed (IL6, IL8, …). Does this mean that the type of intervention does not play a role? Does this mean that all tested biomarkers behave similarly? Maybe the authors can provide a technical control to ascertain that the biomaterial is specifically sensitive to the choosen intervention.

77.       Discussion: the authors have structured their discussion very well and logical in paragraphs to firstly describe what has been published and next, how their data confirm what was known before. But the reader might wonder: what is new to the work presented here? To put things into perspective, please explicitly state what the study adds to the field, maybe at the end of the discussion section in the concluding paragraph. Also, mabe the autors should also stress which part of their findings are, at least partially, in conflict with the data already published on anti-inflammatory therapy and asthma, such as:  mRNA for 8 cytokines are differentially expressed in figure 5 (lung) while only four differentially expressed cytokines are provided in figure 8 (spleen). Is this expected (tissue of origin, mRNA vs protein assay)?

Author Response

Wang and colleagues present a mostly murine-based study on how Sophoraflavone G influences the clincial and molecular phenotype in an asthma mouse model. The autors undetook sereval bioassays to characterise the breathing properties of mice and the immunological phenoytpe of organ compartments. However, even though this is an impressive amount of data, the reviewer feels that the manuscript needs to be improved to make the findings acessible for the readership of The Journal. In detail:     

  1. Introduction: Asthma is characterised by multiple endophenotypes. One distinction is “eosinophilic asthma” vs “non-eosinophilic asthma”. In their introduction, the authors focus on esosinic & Th2-mediated subtype only, for the sake of completeness, please add the information for the readers from the non-asthma-field on how brouad the cinicla spectrum of asthma is on the molecular level as the Int. J. Mol. Sci. is a journal that attracts an audience from several fields.

Responses:

Thank for reviewer’s suggestion. We add more the description of non-eosinophilic asthma in Introduction (Line 56-61) to provide a description of the underlying asthma pattern.

  1. Introductory paragraph on ROS in asthma: appears unlinked to text surrounding it, might be better placed in context (How do ROS and SG interrelate?) or removed.

Responses:

Thank for reviewer’s suggestion. We modified ROS description and placed in the previous paragraph. We also add some compounds from Sophora flavescens to regulate ROS in Introduction.

  1. Figure 1, 2, 3, 4, 5, 6, 7 and 8: please add no of observations (independent technical and biological replicates, i.e. no of measurements, no of litters etc) and figure 9: please add number of technical and biological replicates (passages) for the experiments on BAS2B. This is especially of importance as the current manuscript focusses on “significant is this, hence discussed is this” while some information might lie in those observations for which no significant difference was observed (“the dog that did not bark”). Thus, it is vital to discriminate “not significant because the effect of the intervention is absent or too small to be detected” versus “not significant because the number of independent biological replicates is very small”.

Responses:

Thank for reviewer’s suggestion. All experiments were repeated at least three times.

We added “The data presented as mean±SEM of three independent experiments” in figure legends. Furthermore, BEAS-2B cells used passages 6-8 for cell experiment. (Line 210, line 408-409).

  1. Text affiliated with figures 1 to 8: it would be helpful for the reader to understand the rationale why certain things have been measured (and others have not). Please again keep in mind that readers of The Journal come from different academic backgrounds beyond asthma, so that it is not self-explanatory why IL13 has been investigated for figure 8 while, for instance, IL10 has not. This does not imply that the reviewer thinks data on IL10 needs to be provide, only the authors choice of biomarkers needs to be more transparent throughout the manuscript’s results section. This is important as there are technical reasons for restriction in some cases: individual transcripts or proteins were targeted (researchers choice) versus a pre-build screening set provided by a company was targeted (company’s choice).

Responses:

Thank for reviewer’s suggestion. Scientific experiments should require more people to understand the purpose of the experiment. Therefore, we provide more experimental descriptions in the Results paragraph. We also hope to provide more professional asthma information to assist scholars in other professional fields to understand the experimental purpose in asthma.

  1. Text affiliated with Figure 9: there is certainly a well-thought-out reason why BEAS2B cells were included in this manuscript that entirely deals with murine models before. Please state clearly what the benefit of this experiment is.

Responses:

Inflammatory cytokines or allergens can induce the activation of tracheal epithelial cells, which not only leads to airway inflammation, but these cells then secrete more mucus to obstruct the respiratory tract. Tracheal epithelial cells have an important physical barrier function and can secrete mucus and remove dust and microorganisms from the respiratory tract. However, in the lungs of patients with asthma, immune cells results in the secretion of high amounts of TNF-α to induce tracheal epithelial cells to release excessive chemokines and cytokines, which not only exacerbate and worsen the inflammatory response in the lungs, but also cause remodeling of the airways. Furthermore, goblet cells are epithelial cells that secrete mucus in the respiratory tract. In patients with asthma, allergens or other irritants could stimulate the abnormal proliferation of goblet cells and secrete more mucus, causing breathing difficulties. Hence, we investigate the Sophoraflavanone G regulated inflammatory response in bronchial epithelial cells for evaluated AHR, eosinophil infiltration, goblet cell hyperplasia, and airway inflammation in the lungs of asthmatic mice.

  1. Figure 9: the six subpanels A to F show a very similar appearance: one small bar, two large bars, two smaller bars. This is independent of the intervention (IL10, TNFalpha) and independent of the biomarker anlysed (IL6, IL8, …). Does this mean that the type of intervention does not play a role? Does this mean that all tested biomarkers behave similarly? Maybe the authors can provide a technical control to

Responses:

First, our laboratory screening for anti-inflammatory natural compounds will detect the cell viability in experimental cells. Cytotoxicity of SG detected using the CCK8 assay in BEAS-2B cells. SG did not present significantly cytotoxic effects at a concentration ≤30 μM, and subsequent cell experiments used SG at 0–30 μM.

We usually used 3-4 drug doses to investigate anti-inflammatory responses in cell experiment. Our experiments have found that some natural compounds can

reduce inflammatory response in a dose-dependent manner. However, some drugs can suppress inflammation at low doses and stimulate inflammation at high doses.

We need to adjust the appropriate drug concentration to detect the inflammatory response in tracheal epithelial cells. Therefore, our experimental results often demonstrate that low-dose drug concentrations cannot inhibit the inflammatory response. With the increase of drug concentration, the inflammatory response can be significantly inhibited in inflamed cell experiments

  1. Discussion: the authors have structured their discussion very well and logical in paragraphs to firstly describe what has been published and next, how their data confirm what was known before. But the reader might wonder: what is new to the work presented here? To put things into perspective, please explicitly state what the study adds to the field, maybe at the end of the discussion section in the concluding paragraph. Also, mabe the autors should also stress which part of their findings are, at least partially, in conflict with the data already published on anti-inflammatory therapy and asthma, such as: mRNA for 8 cytokines are differentially expressed in figure 5 (lung) while only four differentially expressed cytokines are provided in figure 8 (spleen). Is this expected (tissue of origin, mRNA vs protein assay)?

Responses:

Thank you again for your positive comments and valuable suggestions to improve the quality of our manuscript. We added more information about inflammatory response in asthma. Abnormal inflammation of tracheal epithelial cells is a risk factor in the development of asthma. Those epithelial cells can release more in cytokines and chemokines to stimulated inflammation of the respiratory tract. Many studies have also pointed out that allergens can stimulate the activation of immune cells in the lungs to cause airway inflammation, airway hyperresponsiveness and oxidative damage in lung of asthmatic patients. Therefore, we assessed local inflammatory and allergic responses and investigated Th2 cell activation in the lungs of asthmatic mice. We also evaluate the systemic allergic reaction to modulate cytokine expression in spleen cells and specific antibody in serum. We will provide a comprehensive discussion in discussion and conclusion paragraph.
